# Lifitegrast Ophthalmic Solution 5% Is a Safe and Efficient Eyedrop for Dry Eye Disease: A Systematic Review and Meta-Analysis

**DOI:** 10.3390/jcm11175014

**Published:** 2022-08-26

**Authors:** Jing-Xing Li, Yi-Yu Tsai, Chun-Ting Lai, You-Ling Li, Ying-Hsuen Wu, Chun-Chi Chiang

**Affiliations:** 1Department of General Medicine, China Medical University Hospital, Taichung 404327, Taiwan; 2School of Medicine, China Medical University, Taichung 406040, Taiwan; 3Institute of Clinical Laboratory Sciences and Medical Biotechnology, National Taiwan University, Taipei 100233, Taiwan; 4Department of Ophthalmology, China Medical University Hospital, China Medical University, Taichung 404327, Taiwan; 5Department of Optometry, Asia University, Taichung 413305, Taiwan; 6School of Medicine, National Taiwan University, Taipei 100233, Taiwan

**Keywords:** dry eye disease, DED, lifitegrast, lymphocyte function-associated antigen 1, LFA-1, Xiidra

## Abstract

Dry eye disease (DED) is a multifactorial disease that causes ocular discomfort and visual impairment on a damaged ocular surface. Lifitegrast, a novel T-cell integrin antagonist, was approved in the United States in July 2016 as a 5% (50 mg/mL) ophthalmic solution for DED management. Currently, no meta-analysis and systemic review based on relevant studies have been conducted. This study aimed to evaluate the efficacy and safety of lifitegrast in patients with DED. We systematically searched Embase, Medline, PubMed, and Web of Science for randomized controlled trials (RCTs) and nonrandomized studies evaluating lifitegrast effects on symptomatic DED. Then, inferior corneal staining score, total corneal staining score (TCSS), nasal lissamine staining score (NLSS), total lissamine staining score, ocular discomfort score (ODS), eye discomfort score (visual analog scale (VAS) score), eye dryness score (EDS), ocular surface disease index score (OSDI-S), and tear break-up time (TBUT) were assessed. Clinical global impression and safety profiles were also evaluated. The studies were pooled in a random-effects model. We included five RCTs, one case–control study, and four longitudinal or retrospective studies, comprising 3197 participants. In the meta-analysis, lifitegrast was superior to the placebo because it improved TCSS, NLSS, TBUT, ODS, eye discomfort score, EDS, and OSDI-Sin DED. However, lifitegrast showed higher risks for ocular and non-ocular treatment-emergent adverse events (TEAEs) overall or at a mild or moderate level. Nonetheless, its incidence of adverse events slightly differed from that in the placebo, especially instillation site discomforts and dysgeusia, thereby considered safe and tolerable. Claims of withdrawal during follow-up caused by TEAEs were extremely rare. Lifitegrast improves DED, although dysgeusia, installation site pain, and irritation may be a concern for some. Overall, most of the adverse events are tolerable. Lifitegrast can alleviate refractory DED and improves patients’ quality of life.

## 1. Introduction

Dry eye disease (DED) symptoms may seriously impair a patient’s quality of life. They may be complex in origin and driven by multiple factors, which can be divided into insufficient tear production and excessive tear evaporation. Although the pathogenesis is still insufficiently understood, DED is associated with ocular surface inflammation [1]. Proinflammatory cytokine IL-17 expression was detected early on the ocular surface and tear film during the development of DED [2,3]. This disease affects the lacrimal glands, eyelids, ocular surface, and interconnecting neural reflex loop, leading to discomfort, visual disturbance, and damage to ocular surface structure and function. DEDs have several well-identified risk factors, such as aging, female (especially menopausal and perimenopausal women [4]), autoimmune diseases [5] (e.g., Sjogren’s syndrome, rheumatoid arthritis, and systemic lupus erythematosus), air pollution [6], blepharitis, systemic agents that decrease tear secretion (e.g., antihistamine and β-blocker), and certain preservatives in topical medications [7].

Integrins are transmembrane receptors that facilitate cell–cell interaction. When lymphocyte function-associated antigen-1 (LFA-1) integrins (also known as CD11a/CD18 or αLβ,2) bind to intercellular adhesion molecule-1 (ICAM-1), it will activate the helper T cells, resulting in an inflammatory cascade. In an animal model of DED, integrin antagonist improved corneal staining and inflammation of the ocular surface [6]. The integrin antagonist blocks T cell recruitment and activation, thereby lessening the inflammatory response. In 2016, the US Food and Drug Administration (FDA) first approved lifitegrast ophthalmic solution 5.0% (Xiidra), a sterile, preservative-free, small-molecule LFA-1 antagonist used for managing DED signs and symptoms. The signs (phase 2 trial [8] and OPUS-1 [9] phase 3 trial) and symptoms (phase 2 trial [8], OPUS-2 trial [10], and OPUS-3 [11] phase 3 trials) of patients with DED taking lifitegrast have shown statistically significant improvements. However, these clinical trials include a particularly controlled patient population potentially not reflected in clinical scenarios. Only two post-marketing studies based on real-world practice have been published. In addition, the measurement of outcomes differed among the research, making it impossible to assess the efficacy and safety profiles of lifitegrast. Refractory DED can be extremely bothersome and seriously reduce the quality of life; alternate treatment is recommended if efficacy and safety are acceptable. Therefore, this study aimed to systemically review the efficacy and safety of lifitegrast in published studies and assess reported adverse events. 

## 2. Methods

We conducted a systematic review and meta-analysis of randomized controlled trials (RCTs) and observational studies (including case–control, prospective, and retrospective cohort studies) regarding the clinical efficacy and safety of lifitegrast. The present study follows the guidelines of Preferred Reporting Items for Systematic Reviews and Meta-analyses (PRISMA) and Meta-analysis of Observational Studies in Epidemiology (MOOSE). 

### 2.1. Literature Search

On 25 June 2022, we searched multiple electronic databases, namely, Ovid PubMed (96), Ovid MEDLINE (39), Ovid EMBASE (119), Ovid Cochrane library (16), and Web of Science and Scopus (77), with no geographic restrictions. The search strategy was designed and conducted by an JXL and YYT, using keywords of (Lifitegrast OR Xiidra OR “lymphocyte function-associated antigen 1” OR LFA-1) AND (“dry eye disease” OR DED OR “dry eye syndrome” OR “meibomian gland dysfunction” OR MGD).

### 2.2. Study Selection

The inclusion criteria were as follows: (1) RCTs, case–control, or cohort studies examining the efficacy and safety of lifitegrast ophthalmic solution 5%, with qualitative data on outcome measurement; (2) the study participants were diagnosed with DED or meibomian gland dysfunction (MGD); and (3) the case group was composed of patients treated with lifitegrast, whereas the control group consisted of individuals treated with a vehicle, other topical medication, or appropriate management. Two authors (Jing-Xing Li and You-Ling Li) independently screened the search results and then scanned the titles and abstracts of citations to exclude studies that did not contain the topic of interest. We checked the full text of potentially eligible studies and included those that met the inclusion criteria. Any discrepancy in article selection was resolved by consulting another author (Chun-Chi Chiang). Afterward, the reference lists of all relevant articles were recursively searched to identify additional studies. Finally, conference proceedings from the American Academy of Ophthalmology between 2011 and 2022 were manually searched to identify additional studies published only in abstract form. In contrast, we excluded the following studies from the meta-analysis: single-center studies, animal studies, post hoc analyses, and studies with insufficient data. Although these studies were excluded from the meta-analysis, their key findings will be included in the following systematic review. 

### 2.3. Data Extraction and Risk-of-Bias Assessment

We extracted the following data from the included studies: first author, year of publication, number of subjects, age, female sex proportion, country, duration of study, study design, and definition of outcomes. The risk of bias of randomized and nonrandomized studies was assessed using version 2 of the Cochrane risk-of-bias tool for randomized trials (RoB 2) and the risk of bias in nonrandomized studies (ROBINS), respectively. For the included randomized studies, the following five domains were evaluated: bias arising from the randomization process, bias caused by deviations from intended interventions, bias resulting from missing outcome data, bias in outcome measurement, and bias in the selection of reported results. For the included nonrandomized studies, the following seven domains were evaluated: risk of bias caused by confounding, risk of bias arising from exposure measurement, risk of bias in participant selection, risk of bias caused by post-exposure interventions, risk of bias resulting from missing data, risk of bias arising from outcome measurement, and risk of bias in the selection of reported results.

### 2.4. Statistical Analysis

All analyses were conducted using the Comprehensive Meta-analysis (CMA) version 3 (Biostat Inc., Englewood, NJ 07631, USA). We calculated the ratio of mean difference (ROMD) with 95% confidence interval (CI) for efficacy measurement by dividing mean change by the grading scale of each outcome. The definition of the mean change is the mean value of the outcome at the endpoint minus baseline in the lifitegrast group. In contrast to the lifitegrast group, the comparison is the control group, including vehicle, thermal pulsation procedure (TPP) and ophthalmic cyclosporine. For safety measurement, pooled odds ratio (OR) with 95% CI was examined. Data on the safety profile can only be extracted from OPUS-1, OPUS-2, OPUS-3 and SONATA studies, which all compare lifitegrast and vehicle. Therefore, the comparison was defined as the placebo group. The statistical heterogeneity across the included studies was evaluated using the *I*^2^ statistic. An *I*^2^ value greater than 50% was considered a substantial heterogeneity. Given that we anticipated clinical heterogeneity, the random-effects model for the meta-analyses was adopted.

## 3. Results

### 3.1. Characteristic of Included Studies

Figure 1 shows the PRISMA study flowchart. Our search identified 198 records after we removed duplicates. However, after scanning the titles and abstracts, we excluded 179 citations. Finally, after examining the full text, our study included 5 RCTs, 1 case–control study, and 4 retrospective studies, with a total of 3197 study participants [9,10,11,12,13,14,15,16,17,18]. Table 1 lists the characteristics of the included studies.

#### Risk of Bias of Included Studies

Figure 2 summarizes the risk of bias in the included randomized and nonrandomized studies. Of the five included RCTs, one was rated with an unclear risk arising from the randomization process, mainly because it was a single-masked randomized study; the four remaining RCTs were double-masked randomized studies. For nonrandomized studies, we rated the studies by De Paz et al. [19] and Epitropoulos et al. [17] at high risk for confounding bias because confounding factors were not identified and properly controlled. Moreover, the study by De Paz et al. [19] showed a high risk of bias emerging from exposure measurement and participant selection because of loose inclusion criteria, no exclusion information, uneven sex distribution, and a very limited number of participants. Meanwhile, the study by Epitropoulos et al. [17] demonstrated an unclear risk of bias arising from exposure measurement because the outcome was only defined according to a subjective report, such as the visual analog scale (VAS) score. Regarding the risk of bias caused by participant selection, studies of De Paz et al. [13] and Hovanesian et al. [18] showed a low risk of bias because of the poor definition of exclusion criteria and no elucidation of inclusion and exclusion criteria, respectively. The study by Hovanesian et al. [18] also demonstrated an unclear risk of bias resulting from missing data because the length of the study spans a year and more than 10% of the participants discontinued lifitegrast at the last administration period. Moreover, we rated the studies by De Paz et al. [19] and Epitropoulos et al. [17] with unclear risk of bias caused by outcome measurement because they merely based it on one subjective scale (VAS and ocular surface disease index [OSDI], respectively). Likewise, the study by Tong et al. [15] showed an unclear risk of bias emerging from outcome measurement because of inconsistent and unreasonable changes in VAS compared with other studies.

### 3.2. Efficacy Outcomes

#### 3.2.1. Objective Evaluation

The meta-analysis revealed that lifitegrast improved the total corneal staining score (TCSS) in patients with DED (ROMD, −0.183; 95% CI, −0.311 to −0.054; *p* = 0.005) (Figure 3B), whereas the inferior corneal staining score (ICSS) did not significantly improve (Figure 3A). Concerning the nasal lissamine staining score (NLSS), the pooled analysis favored lifitegrast over baseline (ROMD, −0.062; 95% CI, −0.078 to −0.047; *p* < 0.001) (Figure 3C). Tear break-up time (TBUT) significantly increased with the use of lifitegrast compared with the baseline (ROMD, 0.259; 95% CI, 0.138–0.379; *p* < 0.001) (Figure 3D).

#### 3.2.2. Subjective Evaluation

The ocular discomfort score (ODS) was improved by lifitegrast use compared with the baseline (ROMD, −0.199; 95% CI, −0.289 to −0.109; *p* < 0.001) (Figure 3E). Lifitegrast also improved the eye discomfort score (ROMD, −0.550; 95% CI, −0.579 to −0.170; *p* < 0.001) (Figure 3F) and the eye dryness score (EDS) (ROMD, −0.371; 95% CI −0.399 to −0.343; *p* < 0.001) (Figure 3G). Concerning the OSDI score, the pooled analysis favored the lifitegrast as well (ROMD, −0.140; 95% CI, −0.331 to −0.052; *p* < 0.001) (Figure 3H).

### 3.3. Safety Outcomes

Regarding the safety profile, the lifitegrast group had more participants reporting more than one ocular treatment-emergent adverse event (TEAE) than the placebo group (OR, 3.250; 95% CI, 2.511–4.025; *p* < 0.001) (Figure 4A). Mild and moderate TEAEs were more observed in the lifitegrast group than in the placebo group (Figure 4B,C), with the risk of moderate TEAE lower than the risk of mild TEAE (OR, 3.091 and 2.395, respectively; both *p* < 0.001). In terms of severe TEAE, the risk was similar between groups (Figure 4D). Approximately 45% of the participants reported at least one ocular TEAE (risk ratio (RR), 3.10; 95% CI, 2.58–3.72; *p* < 0.001). The most frequently reported TEAE in patients treated with lifitegrast was instillation site pain (21.5%), followed by instillation site irritation (15.9%) and instillation site reaction (12.1%) (Table 2). In addition, visual acuity decrease was rare (6.3%) in the pooled analysis, and the risk was not significant compared with that in the placebo group (*p* = 0.559) (Table 2). Compared to the placebo group, the risk of instillation site irritation, pain, reaction, and pruritis, as well as nervous system disorders and dysgeusia, were significantly increased in the lifitegrast group (Table 2). The lifitegrast group had higher odds of developing dysgeusia by approximately 36-fold than the placebo group (RR, 36.06; 95% CI, 13.28–97.88, *p* < 0.001).

As illustrated in Figure 4E, the pooled analysis revealed a significant association of more than one non-ocular TEAE (OR, 4.028; 95% CI, 1.440–11.263). Mild and moderate TEAEs were more common in the lifitegrast group than in the placebo group (Figure 4F,G). The risk of moderate TEAEs was similar to that of mild TEAE (OR, 3.205 and 3.303; *p* = 0.049 and 0.006, respectively), but severe TEAEs showed no difference between the two groups (Figure 4H). Approximately 30.3% of the participants reported at least one non-ocular TEAE, and the most frequently reported were nervous system disorders (17.5%) and dysgeusia (15.0%) (Table 2).

## 4. Discussion

To our best knowledge, this study is the first meta-analysis to examine and summarize the efficacy and safety profile of lifitegrast among patients with DED. Our pooled analysis consisted of not only phase 2 or 3 clinical trials but also phase 4 post-marketing real-world studies. Randomized and nonrandomized studies were all included after screening and eligibility checking. 

Phase 2 study [8] and OPUS-1 [9] study enrolled patients with mild-to-moderate DED, whereas OPUS-2 [10] and OPUS-3 [11] enrolled patients with moderate-to-severe disease. According to the results of post hoc analyses, Holland et al. [20] suggested that the differences in baseline disease severity in the study populations may have caused the various patterns of findings between the phase 2, OPUS-1, and OPUS-2 studies. Lifitegrast improved DED signs in patients with mild-to-moderate disease (phase 2 and OPUS-1 studies) and improved DED symptoms in patients with moderate-to-severe disease (OPUS-2 study). Post hoc analysis of OPUS-2 and OPUS-3 trials stratified participants into four subgroups according to the severity of ICSS and EDS. The subgroup with an ICSS above 1.5 and an EDS of at least 60 at baseline (defined as moderate-to-severe DED) demonstrated a twofold-higher odds of achieving significant improvement [21]. In a retrospective study, 6 month treatment with lifitegrast significantly improved 56% of patients with moderate-to-severe symptomatic DED and moderately improved 36% of them [22]. The most reported adverse event was dysgeusia (16%), consistent with previous findings. Lifitegrast causes a unique metallic or salty taste that may last 3–4 h and annoy patients. Additionally, lifitegrast is convenient, safe and well tolerated over 1 year of use [18].

Several preliminary clinical trials have been conducted on the topical ophthalmic solution SAR 1118, the former name of lifitegrast. A phase 1 study by Semba et al. [22] assessed the safety and tolerability of four escalating concentrations (0.1%, 0.3%, 1.0%, and 5.0%) of lifitegrast with different frequencies of dosing and found that 5% solution given thrice daily is safe and well tolerated. Furthermore, Paskowitz et al. [23] reported that SAR 1118 has favorable pharmacokinetics for therapeutic use. A phase 2 randomized, double-masked, placebo-controlled study conducted by Semba et al. [8] investigated SAR 1118 (0.1%, 1.0%, and 5.0%) on ICSS and OSDI, which improved after receiving 1.0% and 5.0% solutions for 84 days. Adverse events increased as the SAR 1118 dose levels increased, but no severe ocular adverse events were described. In our pooled analysis, only 83 (6.75%) out of 1297 patients could not endure TEAEs and asked for withdrawal. Phase 2, OPUS serial, and SONATA studies did not report actual reasons for withdrawal, but Donnenfeld et al. [12] reported that the most frequent TEAE leading to withdrawal was dysgeusia (1.8%), instillation site reaction (1.8%), increase lacrimation (0.9%), and visual acuity decrease (0.9%). Dysgeusia was not measured and reported in the phase 2 study [8]. Nevertheless, no significant difference was observed between the lifitegrast and placebo.

With regard to DED inflammatory cycle, DED can be initiated or exacerbated by multiple factors that lead to tear instability and changes in tear composition. Stress signaling pathways in the ocular surface cells are then activated, resulting in the production of inflammatory mediators. These mediators recruit and activate CD4 T cells producing cytokines, which cause conjunctival, corneal, and lacrimal gland epithelial disease [24]. Inflammation has a central role in DED pathogenesis and destabilizes the osmolarity of the tear film. During the process of ocular inflammation, surface proteins recruit and further activate T cells. Autoantigens will be released once ocular surface cells are disrupted, followed by cytokine upregulation in the damaged ocular surface. Cellular signaling molecules will up-regulate ICAM-1 expression in cells. Subsequently, ICAM-1 binds LFA-1 integrins on the T-cell surface, finally resulting in T-cell activation [25]. T-cell activation triggers the production of proinflammatory cytokines such as tumor necrosis factor-α (TNF-α), interleukins (ILs; IL-1 and IL-6), and matrix metalloproteinase-9 (MMP-9) [13].

Meibomian gland dysfunction is the most common etiology of DED. Conventional treatments for DED include artificial tear, warm compression, topical anti-inflammatory medication, and antibiotics. Nevertheless, it is frustrating to achieve long-term relief of signs and symptoms. Intense pulsed light (IPL) has been recently introduced in the ophthalmic practice to manage DED originating from MGD [26]. The addition of TPP to IPL reportedly reduces the total number of IPL sessions required and has significant symptomatic benefits after the first session [27]. Patients with MGD have increased levels of MMP-9, one of the proinflammatory mediators, indicating an active inflammatory process [1]. Approximately two-thirds of patients with DED have MGD [28]. TPP simultaneously produces heat to inner eyelids surface and pulsating pressure to the outer eyelids, leading to the evacuation of meibomian gland content. It can improve gland function in MGD and symptoms of DED. However, treatment with lifitegrast showed greater improvement in eye dryness, corneal staining, and eyelid redness than TPP [14]. Furthermore, TPP did not significantly increase lipid layer thickness in comparison with lifitegrast.

Before lifitegrast was made available on the market, lubricant, ophthalmic antibiotic, ophthalmic cyclosporine (Restasis), ophthalmic corticosteroids, eye insert (Lacrisert), tear-stimulating drugs, and autologous blood serum drops were regarded as the conservative treatment approaches for DED. Ophthalmic topical agents aimed to suppress inflammation, but slow improvement in symptoms is often documented. Therefore, concomitant treatment with artificial tears or steroids should be considered. Epitropoulos et al. [17] indicated that lifitegrast was as effective as cyclosporine added to lipid-layer artificial tears for patients with DED who were at risk of failing or refractory to immunomodulatory therapy. Cyclosporine ophthalmic emulsion 0.05% was the first medication approved by FDA for DED in 2003. Cyclosporine interferes with the activity and growth of T cells to block the inflammatory response. However, no studies have analyzed the efficacy of Restasis versus Xiidra, and clinical experience varies in both agents. Xiidra can begin reducing dryness in the eyes within two weeks, whereas Restasis takes up to three months. This is attributed to a different mechanism of action. Restasis inhibits the protein phosphatase and calcineurin, which leads to inhibition of IL-2 production and inhibition of T-cell activation [29]. Nevertheless, Restasis does not suppress activated T cells, whereas Xiidra inhibits the activation of all T-cells by acting directly on the LFA-1/ICAM-1 signaling pathway [30]. Hence, the improvement of DED may become apparent after activated T cells undergo apoptosis in approximately 160 days [31]. Further research comparing the efficacy of lifitegrast with that of ophthalmic cyclosporine may be considered. 

### Strengths and Limitations

The specific strength of this study is a detailed meta-analysis with subgroup analysis to ascertain the efficacy and safety of lifitegrast. However, this study also has certain limitations. First, the outcome measurement varies between the included studies in the objective/subjective scale or number. Therefore, some pooled analyses are merely based on two studies, and some TEAEs may be reported in only one study. Second, although the study by De PaZ et al. [13] showed a high risk of bias in many domains because of a very small sample size, lack of control group, data collection at a single site, and only one measured outcome, the OSDI score data were informative. For further investigation of lifitegrast on OSDI, we included the study for pooled analysis. Third, restricted by available data, we compared the efficacy of lifitegrast with the baseline on various scales after the treatment. If comparing lifitegrast with control from the start of treatment until the end of the study period, scarce data can be used. Fourth, the duration of the included studies ranged from 3 weeks to 12 months, and studies with longer periods of lifitegrast treatment may have greater effects relative to shorter ones. The dose effect of lifitegrast was described based on phase 2, OPUS-1, OPUS-2 and OPUS-3 studies on 14, 42 and 84 days after administration of lifitegrast. Because of the limited number of included studies, additional analysis with stratification of treatment duration is not feasible. Fifth, the relative weight of studies varied in subgroup analyses was attributed to the variation in sample size across the included studies, whereas the overall direction of effects was consistent. Sixth, the severity of DED also varies between included studies, which made it difficult to further analyze the efficacy of lifitegrast with the stratification of severity. Lastly, conducting a pooled analysis on the same endpoint was difficult because the duration of the included study varied from 4 weeks to almost a year.

## 5. Conclusions

After a comprehensive review of several clinical trials, case–control, and longitudinal studies, lifitegrast was confirmed to be a safe topical agent for DED, with promising efficacy. Lifitegrast may provide an alternative treatment option for DED, hence addressing an important need for ophthalmologists who treat this illness. 

## Figures and Tables

**Figure 1 jcm-11-05014-f001:**
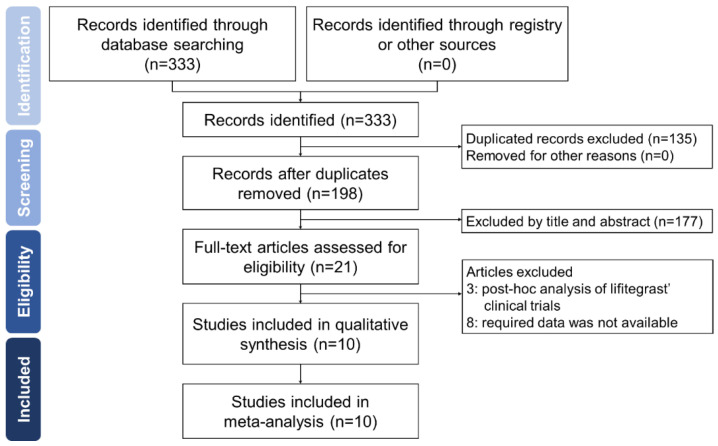
Flow diagram demonstrating the process of study identification.

**Figure 2 jcm-11-05014-f002:**
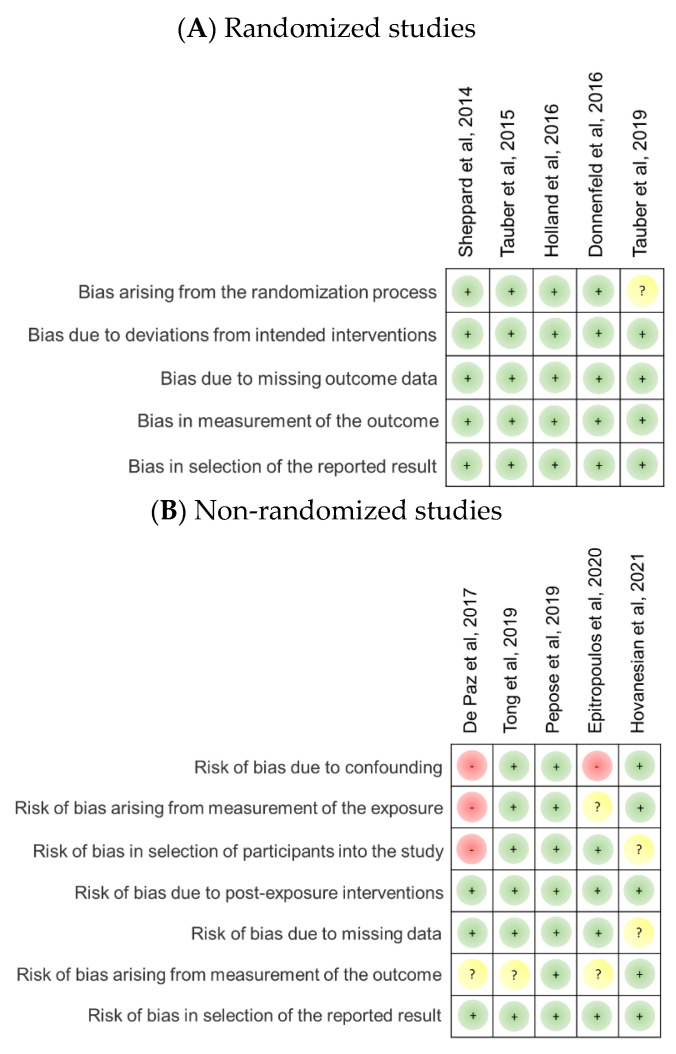
Risk of bias of included studies. (**A**) Risk of bias of included randomized controlled trials. (**B**) Risk of bias of included nonrandomized studies. A green dot denotes the low risk of bias, yellow for unclear risk of bias, and red for high risk of bias [9,10,11,12,13,14,15,16,17,18].

**Figure 3 jcm-11-05014-f003:**
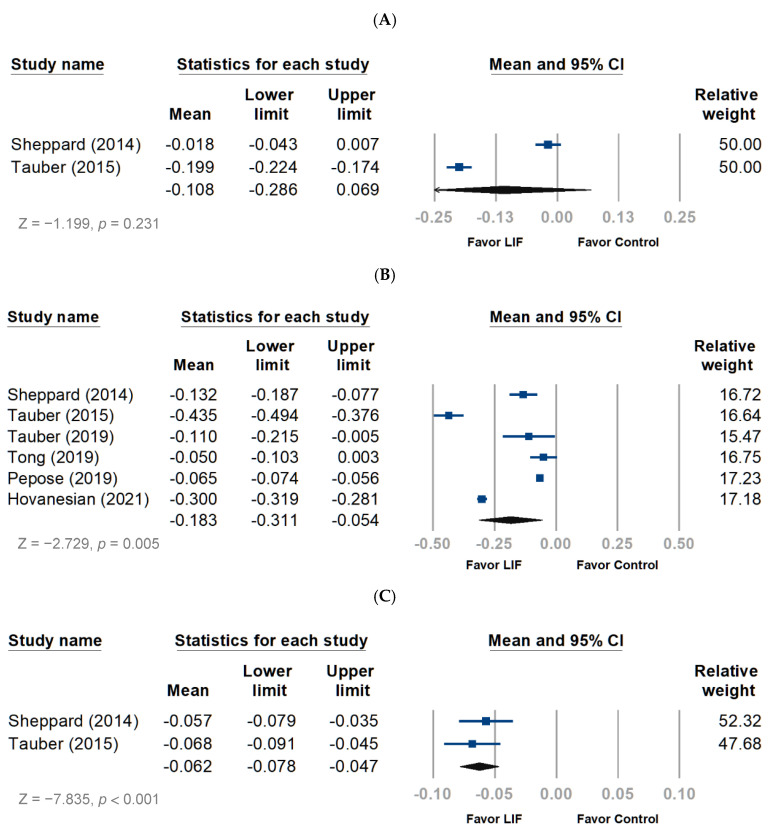
Forest plot of outcome measurement following treatment with lifitegrast. The mean values indicated the ratio of the mean change to each evaluating scale. LIF, lifitegrast. (**A**) Inferior corneal staining score. (**B**) Total corneal staining score. (**C**) Nasal lissamine staining score. (**D**) Tear break-up time. (**E**) Ocular discomfort score. (**F**) Eye discomfort score. (**G**) Eye dryness score. (**H**) Ocular surface disease index score.

**Figure 4 jcm-11-05014-f004:**
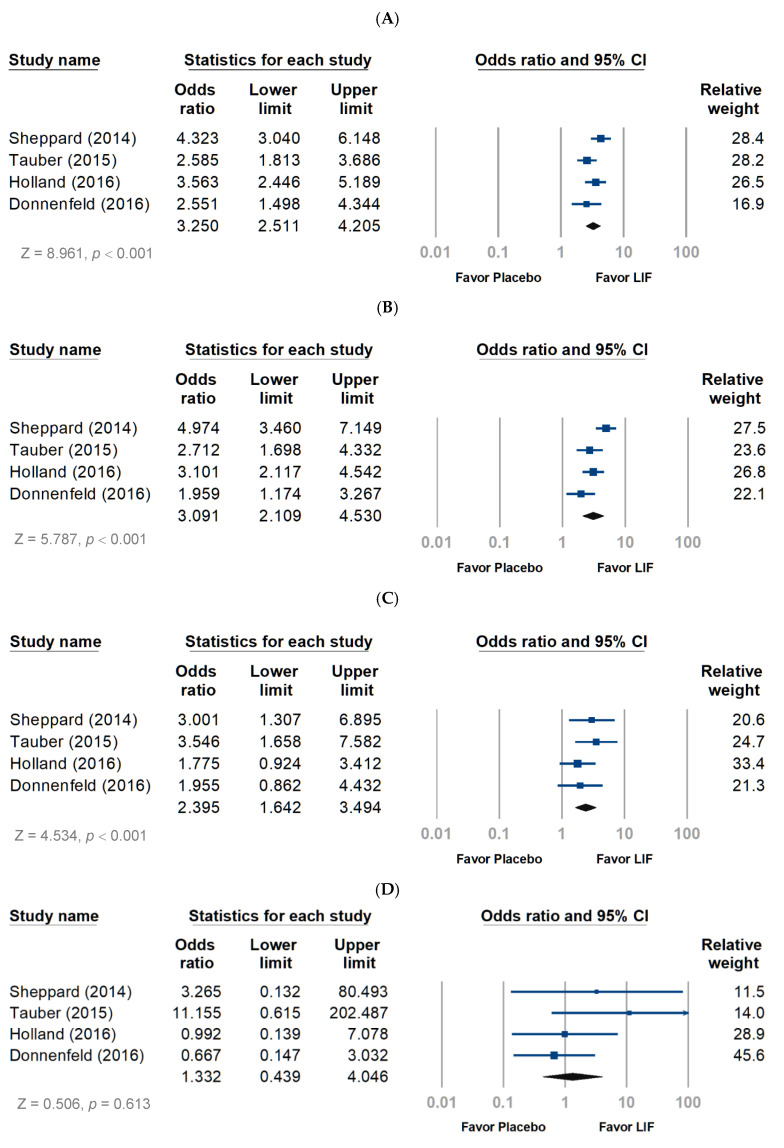
Pooling ocular and non-ocular adverse events (AEs) of lifitegrast treatment. LIF, lifitegrast. (**A**) Ocular AEs, overall. (**B**) Ocular AEs, mild. (**C**) Ocular AEs, moderate. (**D**) Ocular AEs, severe. (**E**) Non-ocular AEs, overall. (**F**) Non-ocular AEs, mild. (**G**) Non-ocular AEs, moderate. (**H**) Non-ocular AEs, severe.

**Table 1 jcm-11-05014-t001:** Overview of included studies.

Author, Year (Study Name)	N	Comparison	Age Mean	Age SD	Female (%)	Country	Duration of Study	Study Design	Blindness	ITT	Outcome Measurement
Sheppard et al., 2014 [9] (OPUS-1)	588	vehicle	60.7	12.0	69.2	US	12 weeks	RCT	double	N/M	CFS, EDS, LGS, ODS, OSDI, TEAE, VAS
Tauber et al., 2015 [10] (OPUS-2)	718	vehicle	58.8	14.1	76.6	US	12 weeks	RCT	double	Yes	CFS, EDS, LGS, ODS, TEAE, VAS
Holland et al., 2016 [11] (OPUS-3)	711	vehicle	58.7	14.5	75.5	US	12 weeks	RCT	double	Yes	EDS, ODS, TEAE, VAS
Donnenfeld et al., 2016 [12] (SONATA)	332	vehicle	59.5	12.7	75.3	US	360 days	RCT	double	N/M	post-instillation comfort, TEAE
De Paz et al., 2017 [13]	14	-	44.9	3.1	85.8	US	4 weeks	case-control	none	N/M	ODSI
Tauber et al., 2019 [14]	50	TPP	65.8	8.9	80.0	US	6 weeks	RCT	single	N/M	BCVA, bulbar conjunctival injection, CFS, lipid layer thickness, meibomian gland patency, MGD score, MMP-9, VAS
Tong et al., 2019 [15]	121	-	60.5	14.4	87.6	US	12 weeks	retrospective cohort	none	N/M	CFS, MGD score, OSDI, TBUT, TEAE
Pepose et al., 2019 [16]	30	-	67.4	-	88.5	US	12 weeks	longitudinal	none	Yes	CFS, MGD score, TBUT, tear osmolality, VAS
Epitropoulos et al., 2020 [17]	33	CYC	69.3	4.2	78.8	US	3 weeks	retrospective cohort	none	N/M	VAS
Hovanesian et al., 2021 [18]	600	-	57.1	-	75.8	US/Canada	12 months	retrospective cohort	none	N/M	CFS, DEQ-5, ODSI, Schirmer score, SPEED, TBUT

CFS, corneal fluorescein staining; CYC, cyclosporine; DEQ-5, 5-item dry eye questionnaire; EDS, Eye dryness score; ITT, intention-to-treat; LGS, lissamine green staining; MGD, meibomian gland disfunction; MMP-9, matrix metalloproteinase-9; N, number of participants; N/M, not mentioned; ODS, ocular discomfort score; OSDI, ocular surface disease index; RCT, randomized controlled trial; SD, standard deviation; SPEED, standardized patient evaluation of eye dryness (SPEED); TBUT, tear break-up time; TEAE, treatment-emergent adverse events; TPP, thermal pulsation procedure; US, United States; VAS, visual analog scale.

**Table 2 jcm-11-05014-t002:** Safety profile of lifitegrast.

Variables	Studies Included	Lifitegrast		Placebo		N	Risk Ratio	95% Cl		*p*
		Event	Total	Event	Total			Lower Limit	Upper Limit	
**Subjects with ≥1 TEAE**	3	504	936	238	824	1760	2.87	2.36	3.50	<0.001
**Subjects with ≥1 ocular TEAE**	4	552	1229	233	1119	2348	3.10	2.58	3.72	<0.001
Instillation site irritation	4	195	1229	33	1119	2348	6.21	4.25	9.06	<0.001
Instillation site pain	1	63	293	11	295	588	7.07	3.64	13.73	<0.001
Instillation site reaction	4	149	1229	37	1119	2348	4.03	2.79	5.84	<0.001
Instillation site pruritus	1	19	293	6	295	588	3.34	1.31	8.49	0.011
Visual acuity reduced	3	55	872	43	765	1637	1.13	0.75	1.71	0.559
Eye pain	1	6	293	5	295	588	1.21	0.37	4.02	0.753
Lacrimation increased	1	7	293	1	295	588	7.20	0.88	58.86	0.066
Eye pruritus	1	5	293	2	295	588	2.54	0.49	13.22	0.267
Ocular hyperemia	1	7	293	4	295	588	1.78	0.52	6.15	0.362
**Subjects with ≥1 non-ocular TEAE**	3	284	936	114	824	1760	2.71	2.13	3.46	<0.001
Nervous system disorders	1	63	359	11	359	588	6.73	3.48	13.01	<0.001
Dysgeusia	3	140	936	4	824	1760	36.06	13.28	97.88	<0.001
**Withdrawal due to ≥1 TEAE**	4	83	1229	25	1119	2348	3.17	2.01	4.99	<0.001

N, number of participants; TEAE, treatment emerging adverse event; 95% Cl, 95% confidence interval.

## Data Availability

Anonymized data of patients are available from the corresponding author on reasonable request.

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
