# Peer review of "Lifitegrast Ophthalmic Solution 5% Is a Safe and Efficient Eyedrop for Dry Eye Disease: A Systematic Review and Meta-Analysis"

_jcm, 2022, doi:10.3390/jcm11175014_

Round 1
Reviewer 1 Report
Outstanding meta-analysis of lifitegrast-associated studies. The manuscript is well-written and the meta-analysis was carried out appropriately. The discussion mentions the study's limitations. Overall, it is an excellent manuscript, which is much needed in the medical literature.
Reviewer 2 Report
Dry Eye Disease (DED)has a multifactorial etiology and different degrees of subjective and objective manifestation. Lifitegrast addresses the inflammatory component of DED.The effects of Lifitegrast are maximum not as single therapy but together with other therapies in order to improve the condition of the ocular surface. An evaluation of some inflammatory mediators (cytokine IL-17)could be an objective recommendation. Analysis of the degree of damage of the ocular surface and the use in optimal proportions of an associated therapy maximise its use. The article highlights the beneficial effects of Lifitegrast with objective improvements in mild and subjective improvements in severe forms of DED, but the associated therapy remains just as important.
Reviewer 3 Report
Thank you for allowing me to review this paper. I consider the topic of this article interesting, but I also think that the article needs a bit more of work.
- In the introduction I miss a clear reason to show why this review is important.
- Is this systematic review registered in PROSPERO? If so, please indicate it
- When the authors claim that the search was done with no “language restrictions”, what does this exactly mean? Was the search done in several languages? In which languages were the articles initially selected published?
- I find a bit confusing the use of conference proceedings for the review but not for the meta-analysis. Also, the authors claim that they search in the main ophthalmology conferences, could you please clarify which ones?
- Could you please clarify which kind of strategies were used in the studies selected? I mean, lifitegrad vs placebo or, for instance, lifitegrad vs artificial tears. It seems that most of the studies used placebo but, according to lines 294 and 295 it also seems that the authors also included studies where TPP was used. In this regard, ‘TPP’ was defined by the 1st time in Table 1, but I would advise doing so also the 1st time that it appears in the main text.
- It would be also interesting knowing if the participants included in the studies had mild, moderate or severe DED. This is in part explained in the discussion, but the information is not easy to find.
- Figure 3 is very informative, but I would appreciate having the units in which each parameter was measured. In this regard, how is obtaining a negative value of corneal staining (A) possible? Similar question related to the tear break up time, how is possible to break the tear before opening the eye? I might be misunderstanding this figure, but I think that it could improve to help to understand the results.
- Also, could you please clarify if all the studies assessed FBUT (fluorescein tear break up time) or used NIBUT (Not invasive tear break up time)?
- Regarding corneal staining, was the criteria to assess it the same in all the studies included? I mean, was the same grading scale used? It would be interesting to explain a little bit how the authors managed to create figure 3.
- Page 8. The authors describe the TEAEs reported more frequently. Are these data related to the control groups or related to the lifitegrad groups? Which ones were more frequent and severe in the lifitegrad group?
- Also, would it be possible clarifying which TEAEs were considered as mid, moderate and severe?
- Table 4. The 1st line contains information related to ‘subjects with ≥ 1 TEAE’ and the 2nd one information related to ‘subjects with ≥ 1 ocular TEAE’. Which ones were considered general TEAE? Also, the columns are not well aligned. I would try to improve the presentation of this table to make the understanding of the information easier.
- Regarding the 1st paragraph of the discussion, I would consider moving the information explaining why the authors did not include the De Paz et al. article (study bias) in other section.
- I think that the 2nd paragraph of P.13 did not discuss any finding, or any information related to this review.
- The dysgeusia reported in control (placebo groups) cannot be related to the lifitegrad. Did the authors investigate which ingredient of the drops could cause this side effect? Was any common ingredient in the drops used in different studies related to this problem?
- In general, although the English is quite good, the reading could improve a little bit. In particular, the discussion section is a bit difficult to follow. Also, some expressions do not sound so natural, for instance where it reads ‘DED can be initiated or exaggerated by…’ I would consider writing ‘DED can be initiated or exacerbated by…’.
- I think that the conclusion is very interesting but, according to the data presented in this paper, I don’t think it is supported by the information presented in this review. For instance, why do the authors consider lifitegrast can be recommended to patients who are are ‘refractory to lubricant or other typical regimens’? In this regard, the conclusion of the present manuscript is the same of the review published in 2019 (Haber et al. 2019). From my point of view, the present article needs to bring something new to be worthy of publication.
Reviewer 4 Report
This is a well-performed study much needed in the field. Currently, there is limited summarizing of the clinical studies on Lifitegrast, with most studies focusing primarily on animal models of DED. Overall, I would like to commend the authors on this manuscript. There are minor changes needed, like multiple fonts found (see line 299) before publishing. Greatly enjoyed reviewing the manuscript.
